# ERAP1 polymorphisms interactions and their association with Behçet's disease susceptibly: Application of Model-Based Multifactor Dimension Reduction Algorithm (MB-MDR)

**Parisa Riahi[1], Anoshirvan Kazemnejad[1]\*, Shayan Mostafaei[2], Akira Meguro [3], Nobuhisa Mizuki[3], Amir Ashraf-Ganjouei[4], Ali Javinani[4], Seyedeh Tahereh Faezi[4], Farhad Shahram[4], Mahdi Mahmoudi [4,5]\***

**1** Department of Biostatistics, Faculty of Medical Sciences, Tarbiat Modares University, Tehran, Iran,
**2** Medical Biology Research Center, Health Technology Institute, Kermanshah University of Medical Sciences, Kermanshah, Iran, **3** Department of Ophthalmology and Visual Science, Yokohama City University Graduate School of Medicine, Yokohama, Japan, **4** Rheumatology Research Center, Tehran University of Medical Sciences, Tehran, Iran, **5** Inflammation Research Center, Tehran University of Medical Sciences, Tehran, Iran

\* mahmoudim@tums.ac.ir (MM); kazem_an@modares.ac.ir (AK)

## Abstract

### Background

Behçet's disease (BD) is a chronic multi-systemic vasculitis with a considerable prevalence in Asian countries. There are many genes associated with a higher risk of developing BD, one of which is *endoplasmic reticulum aminopeptidase-1* (*ERAP1*). In this study, we aimed to investigate the interactions of *ERAP1* single nucleotide polymorphisms (SNPs) using a novel data mining method called Model-based multifactor dimensionality reduction (MB-MDR).

### Methods

We have included 748 BD patients and 776 healthy controls. A peripheral blood sample was collected, and eleven SNPs were assessed. Furthermore, we have applied the MB-MDR method to evaluate the interactions of *ERAP1* gene polymorphisms.

### Results

The TT genotype of rs1065407 had a synergistic effect on BD susceptibility, considering the significant main effect. In the second order of interactions, CC genotype of rs2287987 and GG genotype of rs1065407 had the most prominent synergistic effect (β = 12.74). The mentioned genotypes also had significant interactions with CC genotype of rs26653 and TT genotype of rs30187 in the third-order (β = 12.74 and β = 12.73, respectively).

### Conclusion

To the best of our knowledge, this is the first study investigating the interaction of a particular gene's SNPs in BD patients by applying a novel data mining method. However, future studies investigating the interactions of various genes could clarify this issue.

**Data Availability Statement:** All relevant data are within the paper and its Supporting Information files.

**Funding:** The authors received no specific funding for this work.

**Competing interests:** The authors have declared that no competing interests exist.

# Introduction

Behçet's disease (BD) is a chronic vasculitis presented with multi-systemic signs and symptoms; however, it is majorly separated from other autoimmune diseases by characteristic bipolar aphthosis [1]. With a wide range of prevalence worldwide (from 0.64 per 100,000 in the UK to 420 per 100,000 in Turkey), BD is mostly distributed in countries alongside the Silk Road [2]. According to the considerable prevalence and morbidity of BD in Asian countries, understanding BD's pathophysiology might lead to new therapeutic options and increasing patients' quality of life. Years of research have proven that similar to many other rheumatic disorders, genetic factors have a significant role in BD's course [3].

HLA region has been proven to have a pivotal contribution to the genetic component of BD [4]. BD's association with *HLA-B*51* is proved by several influential studies, including a meta-analysis on 4800 patients that has shown individuals with this allele have an odds ratio of 5.78 for developing BD [5]. In addition to *HLA-B*51*, studies have suggested a link between BD and other genes such as *interleukin 10* (*IL-10*) and *IL-23 receptor* (*IL-23R*), some of which are associated with *HLA-B*51* [6]. In our previous study, we have shown that the *endoplasmic reticulum aminopeptidase-1* (*ERAP1*) gene polymorphisms are associated with *HLA-B*51*, resulting in higher BD susceptibility [7]. ERAP1 is an amino-peptidase responsible for the N-terminal trimming of peptides, which is a critical step in peptides processing and their presentation by MHC-I [8].

Furthermore, ERAP1 takes part in cleaving proinflammatory cytokine receptors such as tumor necrosis factor receptor (TNFR1) from the cell membrane [9]. Polymorphisms of *ERAP1* might alternate the activity of the protein and subsequently changing the structure of peptidome available to HLA-B*51. However, the association of *ERAP1* single nucleotide polymorphisms (SNPs) and BD susceptibility is not entirely clear, and some studies suggest contradictory findings, which need to assess by more comprehensive studies [7, 10, 11].

Up to now, logistic regression for high dimensional and sparse data, parameter estimation is a costly and non-accurate procedure that introduces significant standard errors because sample sizes are too small compared to the order of interaction size. Also, conventional approaches (e.g., logistic regression) used for the analysis of genomic data are oversimplified and usually cannot consider all possible associations between multiple polymorphisms and gene-gene interactions [12]. Multifactor Dimensionality Reduction (MDR) approach is now a reference in the epistasis and SNPs interactions detection field. However, MDR suffers from some significant drawbacks, including that crucial interactions could be missed owing to pooling too many cells together or that proposed MDR analysis will only reveal at most one significant epistasis model, the selection being based on computationally demanding cross-validation and permutation strategies. To overcome the aforementioned hurdles, model-based multifactor dimensionality reduction (MB-MDR) is a flexible framework to detect gene-gene or SNP-SNP interactions. MB-MDR is a non-parametric data mining method that has sufficient power and is capable of investigating the interaction of the unlimited number of genes and polymorphisms [13]. Therefore, we aimed to use the MB-MDR method to identify the interactions of *ERAP1* polymorphisms and their association with BD susceptibly.

# Methods

## Study participants

The present study included 748 BD patients who were referred to the outpatient BD clinic in the Rheumatology Research Center, Shariati Hospital, Tehran, Iran. The International Criteria confirmed patients' diagnosis for Behçet's Disease (ICBD), and patients who were less than 16

**Table 1. Allele frequencies of 11 *ERAP1* SNPs.**

| SNP | Position on chromosome five | Alleles | Amino acid changes | Minor allele frequency, % | | P value | Odd ratio (95% confidence interval) |
|---|---|---|---|---|---|---|---|
| | | | | cases | controls | | |
| rs1065407 | 96,776,379 | T > G | Intronic | 36.6 | 32.5 | 0.018 | 1.20 (1.03–1.39) |
| rs27044 | 96,783,148 | C > G | Glu730Gln | 28.5 | 29.1 | 0.74 | 0.97 (0.83–1.14) |
| rs17482078 | 96,783,162 | C > T | Arg725Gln | 12.6 | 10.3 | 0.052 | 1.25 (1.00–1.56) |
| rs10050860 | 96,786,506 | C > T | Asp575Asn | 12.5 | 10.1 | 0.039 | 1.27 (1.01–1.59) |
| rs30187 | 96,788,627 | C > T | Arg528Lys | 40.1 | 39.7 | 0.82 | 1.02 (0.88–1.18) |
| rs2287987 | 96,793,832 | T > C | Met349Val | 12.5 | 10.2 | 0.040 | 1.27 (1.01–1.59) |
| rs27895 | 96,793,840 | C > T | Gly346Asp | 9.8 | 9.9 | 0.98 | 1.00 (0.79–1.26) |
| rs26618 | 96,795,133 | T > C | Ile276Met | 20.1 | 22.9 | 0.059 | 0.85 (0.71–1.01) |
| rs26653 | 96,803,547 | G > C | Pro127Arg | 40.2 | 39.7 | 0.75 | 1.02 (0.89–1.18) |
| rs3734016 | 96,803,761 | C > T | Glu56Lys | 1.9 | 2.4 | 0.40 | 0.81 (0.50–1.32) |
| rs72773968 | 96,803,892 | G > A | Thr12Ile | 9.8 | 9.9 | 0.88 | 0.98 (0.77–1.25) |

years old or related to each other were excluded from the study [14, 15]. For the control group, we have included 776 healthy individuals with no clinical presentation or family history of any rheumatic disorders or autoimmune diseases, who were matched for sex, age, and ethnicity [16]. Written informed consent was obtained from all individuals themselves or their parents in cases with the age of under 18. The ethical committee of Tehran University of Medical Sciences approved the study protocol, and the relevant university guidelines did all experiments.

## DNA preparation and SNP genotyping

A peripheral blood sample was collected from all participants into EDTA-anticoagulated tubes using venipuncture. Genomic DNA was extracted using the standard phenol/chloroform method, and the extracted DNA samples were stored at −20 ˚C. Approximately 20 ng of the genomic DNA in each sample was used for genotyping. We assessed 10 common missense SNPs from our previous study [7] that were identified in the super-population of the 1000 Genomes project and had a minor allele frequency of more than one percent (*Table 1*). We have also included an intronic SNP (rs1065407) that has been associated with BD in another study [17]. MGB-TaqMan Allelic Discrimination technique was used for SNP genotyping (Applied Biosystems, Foster City, CA, USA). Ten μl of reaction volumes, containing 0.25 μl of distilled water, 4.5 μl of genomic DNA, 0.25 μl of TaqMan genotyping assay mix, and 5 μl of the TaqMan genotyping master mix was used for amplification. The StepOnePlus Real-Time PCR System (Applied Biosystems) and the manufacturer's protocol were used for genotyping the patients and healthy individuals' samples. The allelic call was done using SDS v.1.4 software (Applied Biosystems) and the analysis of allelic discrimination plots. Finally, the genetic makeup of SNPs for each subject was considered as the genotype of that SNP.

## Statistical methods

The continuous variables were indicated as mean ± SD. Allelic and genotypic frequencies of the *ERAP1* SNPs were mentioned as N (%). The genotype distributions of SNPs were tested for deviation from Hardy-Weinberg equilibrium (HWE) in the control group. P-values were corrected for multiple comparisons by the Benjamini-Hochberg approach [18]. Since calculations of the main effect of *ERAP1* SNPs were not available by the model-based multifactor dimensionality reduction (MB-MDR), multiple logistic regression has been used to obtain the main effects of *ERAP1* SNPs, simultaneity. To adjust for main effects, main effects should be

calculated. MB-MDR has been proposed by Calle *et al.* as a dimension reduction method for exploring SNP-SNP interactions with disease susceptibly in case-control association studies [19]. MB-MDR method has proven to be more potent than multifactor dimensionality reduction (MDR) in the presence of genetic heterogeneity [20]. MB-MDR can unify the best of both nonparametric and parametric machine learning algorithms.

On the other hand, characterization, and identification SNP-SNP interactions lack performance in the absence of proper statistical methods and large sample sizes. Logistic regression, as a standard tool for modeling effects and interactions with binary response data, lacks power in the identification of gene interactions in high-order levels due to sparsity and separation [21]. Thus, in this study, SNP-SNP interactions were calculated by the MB-MDR algorithm. MB-MDR shows high power in the presence of all types of noises, such as missing data, genotyping error, genetic heterogeneity, and low sample size [22]. This algorithm was performed by "mbmdr" R package version 3.5.1. To assess the significance in MB-MDR, permutation test with 1000 replications has been done, which corrects for multiple testing (overall marker pairs) and adequately controls the family-wise error rate at $\alpha = 0.05$.

## Results

In this case-control study, 748 patients and 776 age-, sex-, and ethnicity- matched healthy controls were included according to the inclusion and exclusion criteria [16]. In BD patients, the mean age was $40.26 \pm 10.88$ years, and in the control group was $38.88 \pm 11.54$ years (P-value = 0.076). Out of 748 patients and 776 healthy individuals, 448 (59.9%) and 476 (61.3%) were male, respectively (P-value = 0.599). Based on the results of assessing the main effects of *ERAP1* SNPs, the TT genotype of rs1065407 SNP ($\beta$ = 0. 23, and adjusted P-value = 0.034) had a significant synergistic effect on BD. The synergistic effect of an allele is described as the allele increasing the disease risk, and the antagonistic effect is described as the allele having a protective effect regarding the disease susceptibility. In contrast, TT genotype of rs30187 SNP ($\beta$ = -0.26 and adjusted P-value = 0.041) and AA genotype of rs469876 SNP ($\beta$ = -0.20 and adjusted P-value = 0.046) had significant antagonistic effects on BD (*Table 2*). Other *ERAP1* SNPs do not have significant main effects concerning BD susceptibly.

Table 2 summarizes the results of SNP-SNP interactions for six important SNPs (rs1065407, rs30187, rs469876, rs2287987, rs17482078, and rs26653). Based on the results of second-order interaction effects, there were only six significant 2-locus models. For instance, CC genotype of rs2287987 and GG genotype of rs1065407 ($\beta$ = 12.74 and adjusted P-value = $2.12\times10^{-10}$) had a significant synergistic effect on BD susceptibility. rs30187 and rs1065407, CT, and TT genotype ($\beta$ = -0.39 and adjusted P-value of $1.98\times10^{-3}$) had a significant antagonistic effect on BD. Synergistic effects of rs469876 (AA and GG) genotypes with rs1065407 (GG and GT) genotypes were significant as well ($\beta$ = 0.32, adjusted P-value = $4.73\times10^{-3}$). Effects of rs30187 and rs469876 (CC vs. AA) and (TT vs. AG) were also significantly synergistic ($\beta$ = 0.32 adjusted P-value = $2.39\times10^{-2}$). rs26653 (CC) with rs1065407 (GG) had a significant synergistic effect on BD ($\beta$ = 0.76, adjusted P-value = $2.49\times10^{-2}$). However, the results of rs26653 (CT) and rs469876 (AG) showed a significant negative association with BD susceptibly ($\beta$ = -0.42, adjusted P-value = $7.38\times10^{-2}$).

Considering third-order interaction effects, we had five 3-locus models for SNP-SNP interactions of *ERAP1* SNPs. For example, the GG genotype of rs1065407, CC genotype of rs2287987, and CC genotype of rs26653 had a significant synergistic effect on BD by a 3-locus model ($\beta$ = 12.74, adjusted P-value = $2.13\times10^{-10}$). However, the 3-locus model (rs1065407, rs2287987, rs26653) did not have any significant antagonistic effect on BD. Considering rs1065407, rs2287987, and rs30187, results reveal that the synergistic effect of (GG, CC, and

**Table 2. Model-based multifactor dimensionality reduction algorithm for assessing the main and interaction effects of 11 *ERAP1* SNPs on Behçet's disease risk (748 Iranian BD patients and776 healthy individuals).**

| Order | Significant Effects | Synergistic Effect | | | | Antagonism Effect | | | | Permutation Test |
|---|---|---|---|---|---|---|---|---|---|---|
| | | N. levels | Genotypes | Coefficient | Adj. P-value | N. levels | Genotypes | Coefficient | Adj. P-value | Perm. P-value |
| Main Effects | rs1065407 | 1 | TT | 0.23 | 0.034 | 0 | NA | NA | NA | 0.019 |
| | rs30187 | 0 | NA | NA | NA | 1 | TT | -0.26 | 0.041 | 0.18 |
| | rs469876 | 0 | NA | NA | NA | 1 | AA | -0.20 | 0.046 | 0.054 |
| #2 order interactions | rs2287987+rs1065407 | 1 | CC+GG | 12.74 | $2.12\times10^{-10}$ | 0 | NA | NA | NA | 0.065 |
| | rs30187+rs1065407 | 0 | NA | NA | NA | 1 | CT+TT | -0.39 | $1.98\times10^{-3}$ | 0.053 |
| | rs469876+rs1065407 | 2 | AA+GG GG+GT | 0.32 | $4.73\times10^{-3}$ | 0 | NA | NA | NA | 0.181 |
| | rs30187+rs469876 | 2 | CC+AA TT+AG | 0.32 | $2.39\times10^{-2}$ | 0 | NA | NA | NA | 0.091 |
| | rs26653+rs1065407 | 1 | CC+GG | 0.76 | $2.49\times10^{-2}$ | 0 | NA | NA | NA | 0.210 |
| | rs26653+rs469876 | 2 | CC+AA GG+AG | 0.54 | $2.83\times10^{-2}$ | 1 | CT+AG | -0.42 | $7.38\times10^{-2}$ | 0.193 |
| #3 order Interaction | rs1065407+rs2287987 +rs26653 | 1 | GG+CC+ CC | 12.74 | $2.13\times10^{-10}$ | 0 | NA | NA | NA | 0.243 |
| | rs1065407+rs2287987 +rs30187 | 1 | GG+CC+ TT | 12.73 | $2.15\times10^{-10}$ | 1 | TT+ CT+ CT | -0.39 | $5.95\times10^{-2}$ | 0.230 |
| | rs1065407+rs30187 +rs469876 | 3 | GG+TT+AG GT +CT+AA | 0.43 | $2.87\times10^{-2}$ | 1 | TT+ CT+ AG | -0.67 | $1.26\times10^{-3}$ | 0.169 |
| | rs30187+ rs1065407 +rs26653 | 4 | CC+GG+CC TT +GT+GG | 0.77 | $2.36\times10^{-2}$ | 0 | NA | NA | NA | 0.137 |
| | rs1065407+rs2287987 +rs469876 | 2 | GG+CC+GG GT +TT+AA | 0.04 | $9.77\times10^{-1}$ | 1 | TT+CT+AG | -0.92 | $3.18\times10^{-2}$ | 0.229 |
| #4 order Interaction | rs1065407+rs2287987 +rs30187+rs26653 | 7 | GG+CC+ CC+CC GT+CT+ TT+CG | 0.53 | $1.94\times10^{-1}$ | 2 | TT+TT+CT+ GG GT+CT+TT+CG | -0.88 | $7.50\times10^{-3}$ | 0.184 |
| | rs1065407+rs2287987 +rs26653+rs469876 | 5 | GG+CC+ CC+GG GT+CT+ GG+AA | 0.66 | $4.49\times10^{-1}$ | 2 | TT+TT+ CG+AG GT+CT+GG+AG | -0.65 | $1.18\times10^{-2}$ | 0.219 |
| #5 order Interaction | rs1065407+rs2287987 +rs30187+rs26653 +rs17482078 | 11 | GT+TT+ CC+CC + TT TT+CT+TT +CG+ CT | 0.32 | $3.93\times10^{-1}$ | 2 | TT+CT+ CT+GG + CT GT+TT+ TT +CG+ TT | -0.89 | $7.25\times10^{-3}$ | 0.032 |

TT) genotypes and the antagonistic effect of (TT, CT and CT) genotypes on BD, were significant as well. Besides, rs1065407 (TT), rs30187 (CT) and rs469876 (AG) had a significant antagonistic effect on BD (β = -0.67, adjusted P-value = $1.26\times10^{-3}$). In addition, rs1065407 (TT), rs2287987 (CT) and rs469876 (AG) interaction had a significant antagonistic effect on BD (β = -0.92, adjusted P-value = $3.18\times10^{-2}$). In contrast, (rs1065407: GG, rs30187: TT, rs469876: AG), (rs1065407: GG, rs2287987: CC, rs469876: GG), and (rs30187: CC, rs1065407: GG, rs26653: CC) had significant synergistic effects on BD. More details are shown in the third-order interaction section of Table 2.

Results of fourth-order interaction effects indicated that (rs1065407: GG, rs2287987: CC, rs30187: CC, rs26653: CC) and (rs1065407: GG, rs2287987: CC, rs26653: CC, rs469876: GG) had significant synergistic effects on BD. In contrast, (rs1065407: TT, rs2287987: TT, rs30187: CT, rs26653: GG) and (rs1065407: TT, rs2287987: TT, rs26653: CG, rs469876: AG) had significant antagonistic effects on BD. Based on the results of five-order interaction effects, (rs1065407: GT, rs2287987: TT, rs30187: CC, rs26653: CC, rs17482078: TT) had a significant synergistic effect on BD (β = 0.32, adjusted P-value = $3.93\times10^{-1}$). However, (rs1065407: TT, rs2287987: CT, rs30187: CT, rs26653: GG, rs17482078: CT) had a significant antagonistic effect on BD (β = -0.89, adjusted P-value = $7.25\times10^{-3}$). In six- order interaction effects, no significant effects were observed (*Table 2*).

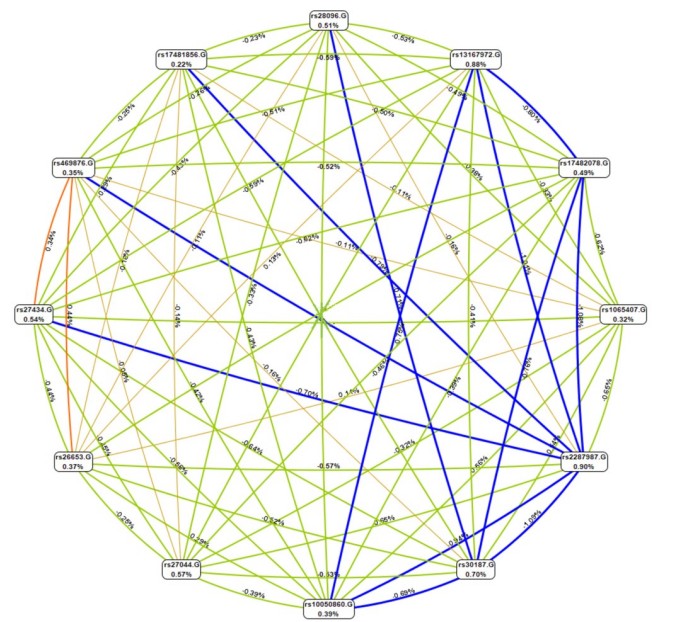

**Fig 1. SNP-SNP entropy-based interaction network of 11 *ERAP1* SNPs in 748 Iranian BD patients.**

More details of the results of 11 *ERAP1* SNP-SNP interactions are presented in the supplementary Table. Also, the entropy-based interaction network of 11 *ERAP1* SNPs was shown in *Fig 1* by using MDR. To assess the sensitivity and cross-validity of the results of MB-MDR, permutation results are shown in the last column of Table 2.

## Discussion

In this study, we aimed to investigate the interactions of the *ERAP1* gene polymorphisms and their associations with BD susceptibility in an Iranian cohort. Using the MB-MDR package, we have found plenty of synergistic and antagonistic significant interactions between *ERAP1* polymorphisms and BD development. Considering the main effects, the TT genotype of rs1065407 had a synergistic effect on BD susceptibility. In the second-order interactions, CC genotype of rs2287987 and GG genotype of rs1065407 had the most prominent synergistic effect (β = 12.74). Furthermore, the mentioned genotypes also had significant interactions with CC genotype of rs26653 and TT genotype of rs30187 in the third-order (β = 12.74 and β = 12.73, respectively). Hence, we propose that the genotypes, as mentioned earlier of rs2287987, rs1065407, rs26653, and rs30187, could have prominent interactions resulting in a higher risk of developing BD.

*ERAP1* gene is located in the 5q15 chromosome, and its expression has been observed in many tissues [23]. There are two main processes that ERAP1 is proposed to have a role in them. First, this amino-peptidase is involved in optimizing the length of peptides to bind with MHC-class I molecules by trimming their N-terminal in the endoplasmic reticulum (ER) [23]. Moreover, ERAP1 is involved in the cleavage process of various cytokine receptors such as TNFR1, Interleukin 1 receptor II (IL-1RII), and Interleukin 6 receptor α (IL-6 α), which results in receptor shedding [24, 25]. Previous studies have shown that the *ERAP1* gene is associated with other autoimmune disorders such as ankylosing spondylitis (AS) and psoriasis [26, 27]. Homozygosity of *ERAP1* polymorphisms is proposed to be correlated with a lower risk of AS and psoriasis, whereas it might be associated with a higher risk of developing BD [28, 29].

These differences could be justified by the fact that loading different peptides on MHC-class I molecules can alter the subsequent immune response.

Our results indicated that the homozygous genotypes of minor alleles of rs2287987, rs1065407, rs26653, and rs30187 had the most prominent interactions causing BD susceptibility. In this regard, it has been demonstrated that the frequencies of the homozygous alleles of the *ERAP1* gene are higher among BD patients [11]. As it was shown in further studies, these combinations of homozygote *ERAP1* SNPs could result in alternations in the surface electrostatic potential of the protein [30]. These changes might alter the trimming activity of ERAP1, resulting in an altered composition of peptidome that is available for binding to HLA-B*51. This claim could support the higher risk of developing BD observed in individuals carrying the mentioned genotypes. Furthermore, some SNPs such as rs30187 (Arg528Lys) are placed proximal to the entrance pocket of the protein [28]. Amino acid changes in such positions could modify the ideal structure of the protein and alter the enzyme activity.

Although several studies have investigated the association of *ERAP1* polymorphisms and BD, there have been some contradictory findings that motivated us to utilize a more complex statistical method for addressing this issue. Zhang *et al*. evaluated 930 Chinese patients and proposed that rs1065407 and rs10050860 might be associated with increased risk of BD [17]. Sousa and colleagues studied another Iranian cohort and proposed that rs10050860 and rs13154629 might contribute to the genetic susceptibility of BD [15]. Moreover, Conde-Jaldón *et al*. found that homozygous genotypes for the minor alleles of rs27044, rs10050860, rs30187, and rs2287987 could be considered as risk factors for BD [10]. Takeuchi and colleagues found a haplotype consisting of 10 SNPs (five of which were non-ancestral), which was associated with a higher risk of developing BD, especially in those individuals who carry *HLA-B*51* [30]. Interestingly, our results indicated that homozygote genotypes of minor alleles of rs30187 and rs2287987 are associated with a higher risk of BD. rs30187 and rs2287987 are among those five SNPs that their non-ancestral alleles were mentioned in Takeuchi's study. Finally, the previous study by our team and the study on the Turkish population revealed that *ERAP1* polymorphisms have epistatic interactions with *HLA-B*51* contributing to BD risk [7, 30].

In conclusion, this is the first study investigating the interaction of a particular gene's SNPs in BD patients by applying a novel data mining method (MB-MDR package). Model-Based MDR as a flexible framework and a reference method to detect gene–gene or SNP-SNP interactions has adequate power even the presence of genotyping errors, missing genotypes, and genetic heterogeneity in this study compare with traditional methods (e.g., logistics regression). Finally, a significant interaction between minor genotypes of *ERAP1* polymorphisms was observed in BD patients in comparison to healthy individuals. rs2287987, rs1065407, rs26653, and rs30187 interactions had the strongest association with developing BD in our study population. Taken together, these findings imply the contribution of *ERAP1* polymorphisms in BD pathogenesis. However, further studies investigating the interactions of different genes could shed more light on this issue.

## Supporting information

**S1 Table. Model-based multifactor dimensionality reduction algorithm for assessing the main and interaction effects of 11 ERAP1 SNPs on Behçet's disease risk (748 Iranian BD patients and 776 healthy individuals).**
(DOCX)

**S1 File.**
(RAR)

## Author Contributions

**Conceptualization:** Parisa Riahi, Anoshirvan Kazemnejad, Mahdi Mahmoudi.

**Data curation:** Parisa Riahi, Amir Ashraf-Ganjouei, Ali Javinani.

**Formal analysis:** Parisa Riahi, Shayan Mostafaei, Akira Meguro, Nobuhisa Mizuki, Amir Ashraf-Ganjouei, Ali Javinani.

**Methodology:** Anoshirvan Kazemnejad, Akira Meguro, Nobuhisa Mizuki.

**Resources:** Shayan Mostafaei, Seyedeh Tahereh Faezi, Farhad Shahram, Mahdi Mahmoudi.

**Supervision:** Anoshirvan Kazemnejad, Seyedeh Tahereh Faezi, Farhad Shahram, Mahdi Mahmoudi.

**Writing – original draft:** Parisa Riahi, Shayan Mostafaei, Amir Ashraf-Ganjouei, Ali Javinani.

**Writing – review & editing:** Parisa Riahi, Anoshirvan Kazemnejad, Shayan Mostafaei, Akira Meguro, Nobuhisa Mizuki, Seyedeh Tahereh Faezi, Farhad Shahram, Mahdi Mahmoudi.

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
